# Evaluation and discussion of handmade face-masks and commercial diving-equipment as personal protection in pandemic scenarios

Mortimer Gierthmuehlen[1]*, Bernd Kuhlenkoetter[2], Yaroslav Parpaley[1], Stephan Gierthmuehlen[3], Dieter Köhler[4], Dominic Dellweg[4]

1 Department of Neurosurgery, Ruhr-University Bochum, Universitaetsklinikum Knappschaftskrankenhaus Bochum GmbH, Bochum, Germany, 2 Production Systems, Faculty of Mechanical Engineering, Ruhr-University Bochum, Bochum, Germany, 3 Medical Law, CausaConcilio, Kiel, Germany, 4 Department of Pulmonology & Intensive Care, Kloster Grafschaft, Schmallenberg, Germany

* mortimer.gierthmuehlen@ruhr-uni-bochum.de

**Data Availability Statement:** All relevant data is within the manuscript.

## Abstract

### Objective

Pandemic scenarios like the current Corona outbreak show the vulnerability of both globalized markets and just-in-time production processes for urgent medical equipment. Even usually cheap personal protection equipment becomes excessively expensive or is not deliverable at all. To avoid dangerous situations especially to medical professionals, but also to affected patients, 3D-printer and maker-communities have teamed up to develop and print shields, masks and adapters to help the medical personnel. In this study, we investigate three home-made respiratory masks for filter and protection efficacy and discuss the results and legal aspects.

### Materials and methods

A home-printed respiratory mask with a commercial filter, a scuba-diving mask with a commercial filter and a mask sewn from a vacuum cleaner bag were investigated with 99mTc-labeled NaCl-aerosol, and the respective filter-efficacy was measured under a scintigraphic camera.

### Results

The sewn mask from a vacuum cleaner bag had a filter efficacy of 69.76%, the 3D-printed mask of 39.27% and the scuba-diving mask of 85.07%.

### Conclusion

Home-printed personal protection equipment can be a–yet less efficient–alternative against aerosol in case professional masks are not available, but legal aspects of their use and distribution have to be kept in mind in order to avoid compensation claims.

**Funding:** We acknowledge support by the DFG Open Access Publication Funds of the Ruhr-University Bochum.

**Competing interests:** MG is founder and consultant of Neuroloop GmbH, which has no link to this manuscript. This does not alter our adherence to PLOS ONE policies on sharing data and materials. The authors therefore have declared that no competing interests exist.

# Introduction

The current Corona pandemic demonstrates the difficulties of modern economies to react sufficiently to the sudden increased demands of everyday consumable supplies. Personal protection equipment (PPE) such as face masks, which usually cost few cents, become extremely rare and are not available over days and weeks, leading to potentially dangerous scenarios especially for medical personnel.

With the increased popularity of 3D printers fast and cheap production of plastic-based items is possible. Strong and enthusiastic maker-communities have formed over the last decade, till now concentrating on making everyday life more pleasant and comfortable. Within the increased demand on PPE, these "maker"-communities have realized their social responsibility and teamed-up to support hospitals and healthcare professionals with computer-designed masks [1], shields [2] and even simple ventilators [3] printed on consumer 3D devices. Beside completely new constructions, some "makers" concentrate on adding printed accessories such as adapters to commercially available devices to make them usable in the healthcare environment, or print spare parts for ventilators which are currently not available. The National Institutes of Health (NIH) installed a public repository for such constructions where users can download the respective files for their 3D-printers [4]. Additionally, the centers of disease control (CDC), tailors and committed amateurs publish sewing-instructions for handmade face-covering masks [5].

We recently evaluated different commercial masks and a scarf with respect to their filter efficacy [6]. As there is a growing number of templates for 3D printed masks on public depositories (e.g. Thingiverse, www.thingiverse.com), we chose one of the most popular templates (highest number of downloads and "likes" to that time) for this study as "Mask 1". There is also growing interest in using diving-masks as a personal protection device [7,8], therefore we decided to include this popular mask in our study as well. The third mask we evaluated was home-sewn from a home-made template, added as a supplementary file. In this study we would like to compare these masks with respect to their efficacy, discuss their use in medical environments and legal aspects during the pandemic scenario.

# Materials and methods

### Aerosol

The method was recently described by our workgroup on Research-Square [6], and Sodium-Chloride (NaCl) aerosol is used to test respiratory masks during their official classification process (EN149:2001). In brief, 99m-Tc-DTPA (diethylenetriamine 131 pentaacetate) with an activity of 150MBq/ml was nebulized using a Pari LC Sprint Star nebulizer (Paris, Starnberg, Germany), filled with 6ml of solution. The output of the nebulizer was set at 360-500mg/min with a (wet) mass median aerodynamic diameter (MMAD) of 2.4–3.3 μm. With drying, we calculated a diameter between 0.58–0,66μm for the NaCl-aerosol, in concordance with the German DIN-classification EN149. For every run the aerosol was delivered continuously for 25 seconds in a sealed plastic chamber (Iris 70-liter, model #135455, IRIS Ohyama, Sendai, Japan, Fig 1) and spread with two 5cm ventilators running for 5s. In the chamber, a human-sized plastic head with silicon covered surface mimicking the skin was installed (Respironics, 95 Murrysville, PA, USA) with the nasal and mouth opening connected to a suction tube on the backside leading out of the box (Fig 2). A second tube left the box without being connected to a head, with the opening free inside the box (Fig 3, black arrow). Each of the two tubes was connected to a filter (Iso-Gard #19212, Teleflex Medical 116 GmbH, Fellbach, Germany) and then attached to two synchronized artificial lungs with two separate bellows (dual adult test lung model 5600i,

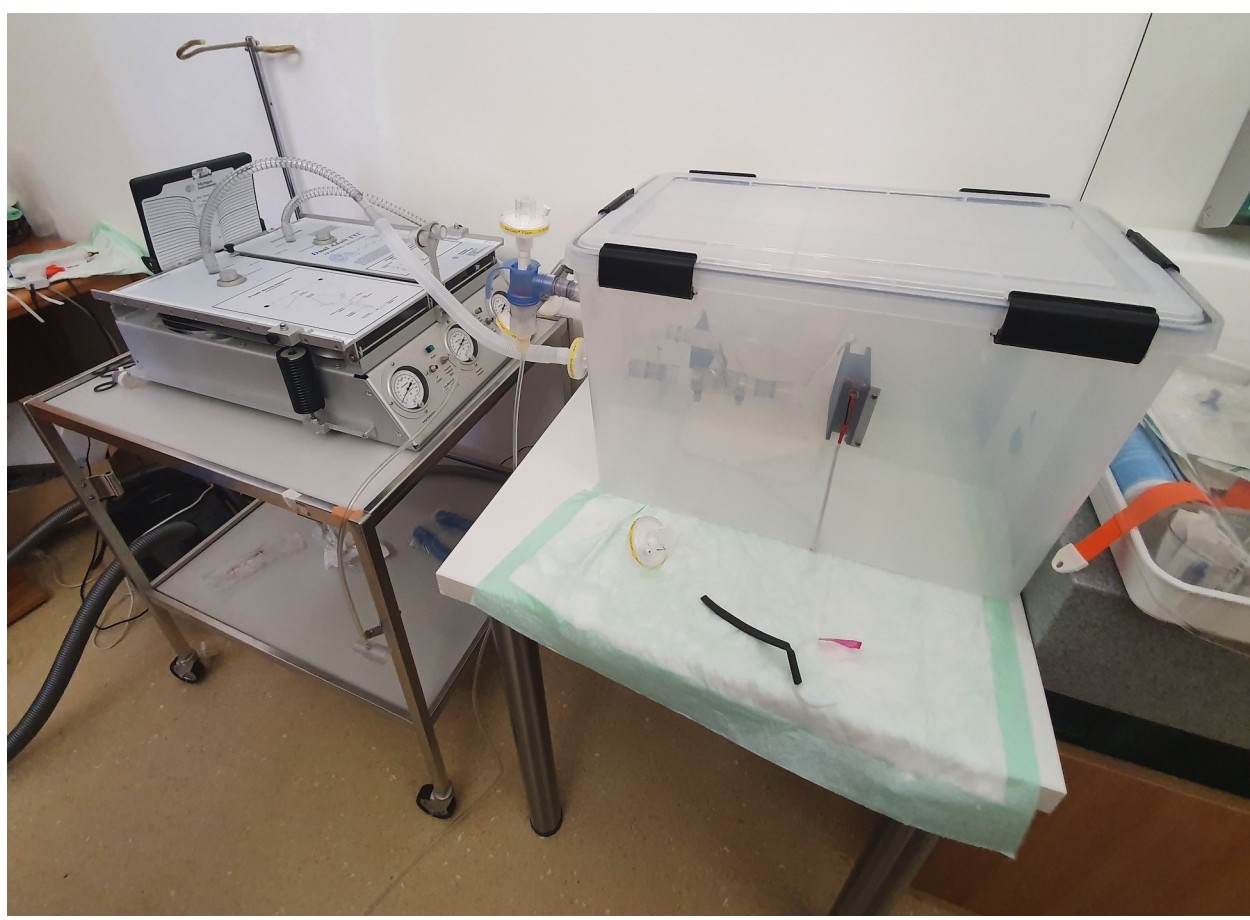

**Fig 1. Test-setup.** Setup of the lungs (left) and the box (right).

Michigan Instruments, 126 Kentwood, MI, USA). As both the test-filter and the reference filter (each time a new one was used) were exposed to $Tc^{99m}$ at the very same time and the entire measurements of all masks took appr. 1h, the natural decay of $Tc^{99m}$ was irrelevant.

## Test run and analysis

The nebulizer delivered the aerosol for 25s into the chamber, then the ventilators distributed it equally over 5s. After another 5s, 10 breaths with 1L were performed over 50s, then the aerosol

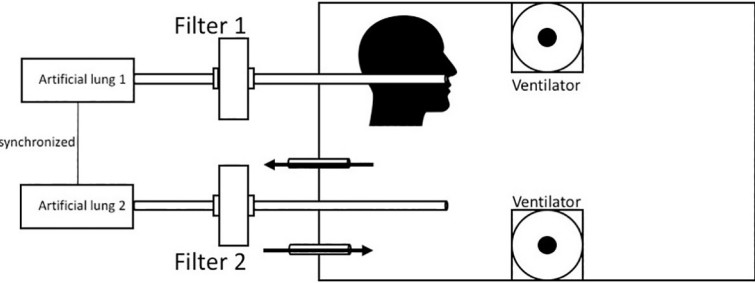

**Fig 2. Schematic setup.** Schematic drawing of the setup seen in Fig 1.

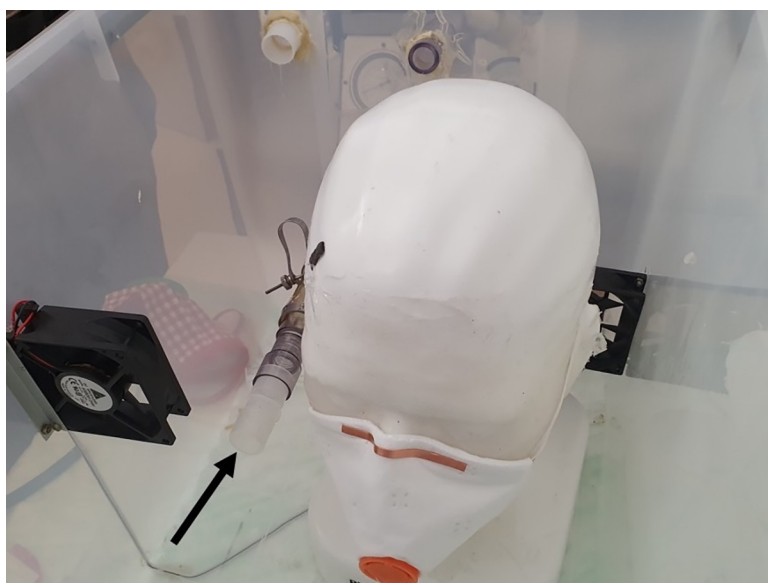

**Fig 3. Measuring box.** Photo from inside of the measuring box with the open pipe (arrow) and the mask attached to the phantom head.

was evacuated with a filtered vacuum. The physiologic respiratory frequency is about 12/min with a tidal volume of 0.5L, but the inspiration volume can be extended to up to 3L. Filter 1 was installed between head and lung, filter 2 was between the freely ending tube in the box and the lung.

After the run, the two filters were placed on a scintigraphy camera (ECAM Scintron, Medical imaging electronics GmbH, Seth, Germany) and the radioactivity was counted for one minute. Regions-Of-Interest (ROI) were placed over each filter and over a neutral position representing background activity. Activity was measured in counts per minute. Background activity was subtracted from filter 1 and filter 2 counts. The ratio filter 1 and filter 2 was calculated representing the efficacy of the applied filtering device in %. The 3D-printed mask was evaluated 4 times, while the home-sewn mask and Easybreath® were both tested 3 times respectively.

## Masks

The first mask (Fig 4A) was sewn from a double-layer microfleece vacuum cleaner bag (McFilter MSM) using a home-made design-template (Fig 4B) and equipped with a home-printed expiration valve (PLA, Primacreator Primavalue, Malmo, Sweden, filament with 1,75mm diameter), printed on an Ender 3 pro Printer (Crealty, Shenzhen, China) with a nozzle-temperature auf 210˚C.

The solid face mask was designed by the maker-community (https://www.thingiverse.com/thing:4225667) and printed with Tefabloc TPE (Verbatim, Charlotte, USA, 1.75mm diameter) on a home-made CoreXY printer and a nozzle-temperature of 235˚C. The initial filter-inlet was replaced by a home-designed adapter (https://cults3d.com/en/3d-printing/covid-19-mask-easy-to-print-no-support-filter-required-inhol) printed with PLA (Verbatim, Charlotte, USA, 1.75mm diameter) on the same home-made printer and a nozzle temperature of 205˚C. Two HME-filters (Iso-Gard #19212, Teleflex Medical 116 GmbH, Fellbach, Germany) were attached to the adapter, two rubber bands kept the mask tightly attached to the head (Figs 5 and 6).

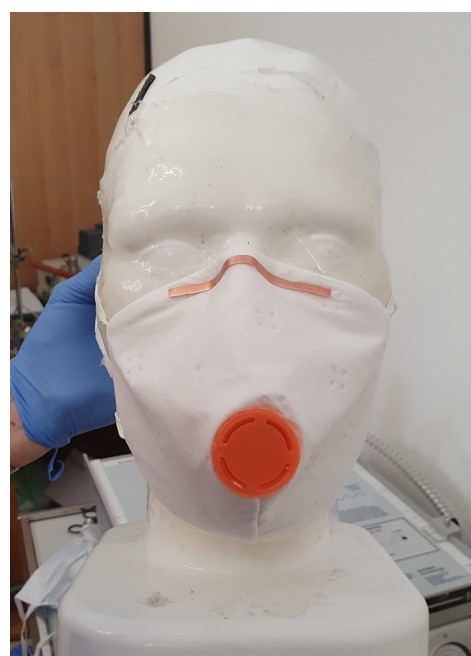 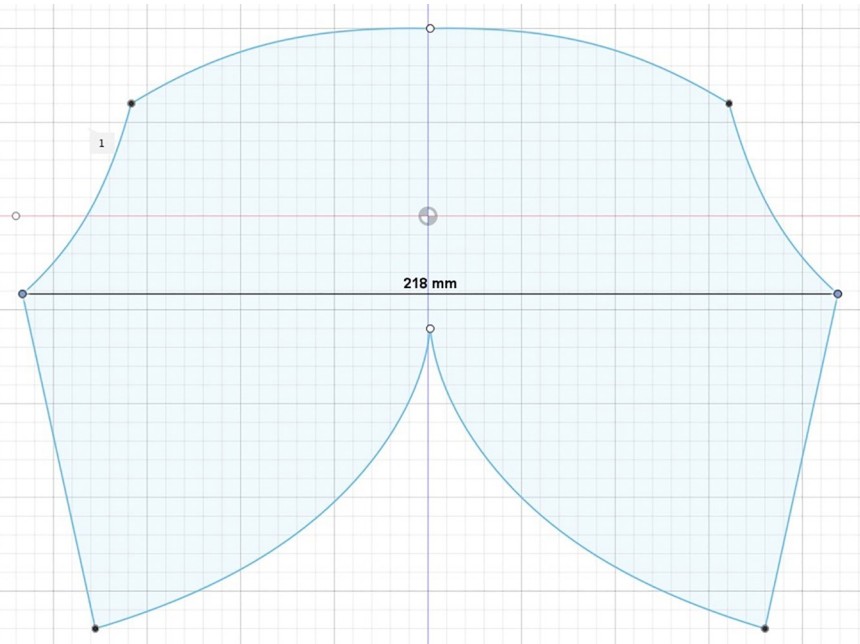

**Fig 4.** **(a)** Home-made mask. Home-made mask sewn from microfleece vacuum cleaner bag with a home-printed expiration valve. **(b)** Template of the home-made mask. A home-designed template inspired by commercial masks was used to cut the microfleece.

A commercial full-face mask for diving purposes (Easybreath®, Decathlon, Villeneuve-d'Ascq, France) was equipped with a community-designed adapter (https://www.thingiverse.com/thing:4269938). The adapter was printed on the same home-made printer with PlA

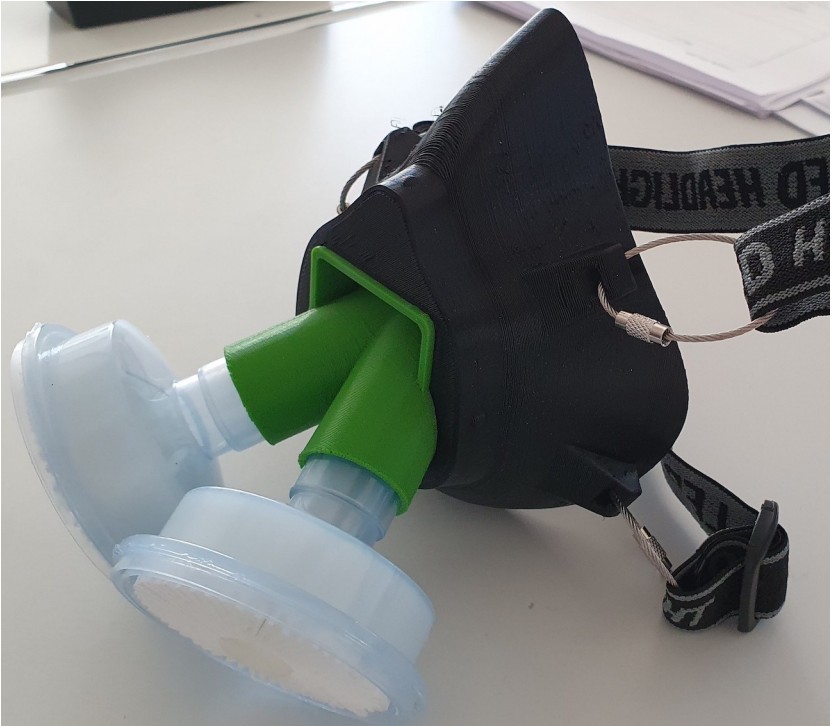

**Fig 5. 3D-printed mask.** 3D-printed mask from TPE with a printed PLA adapter and two attached filters.

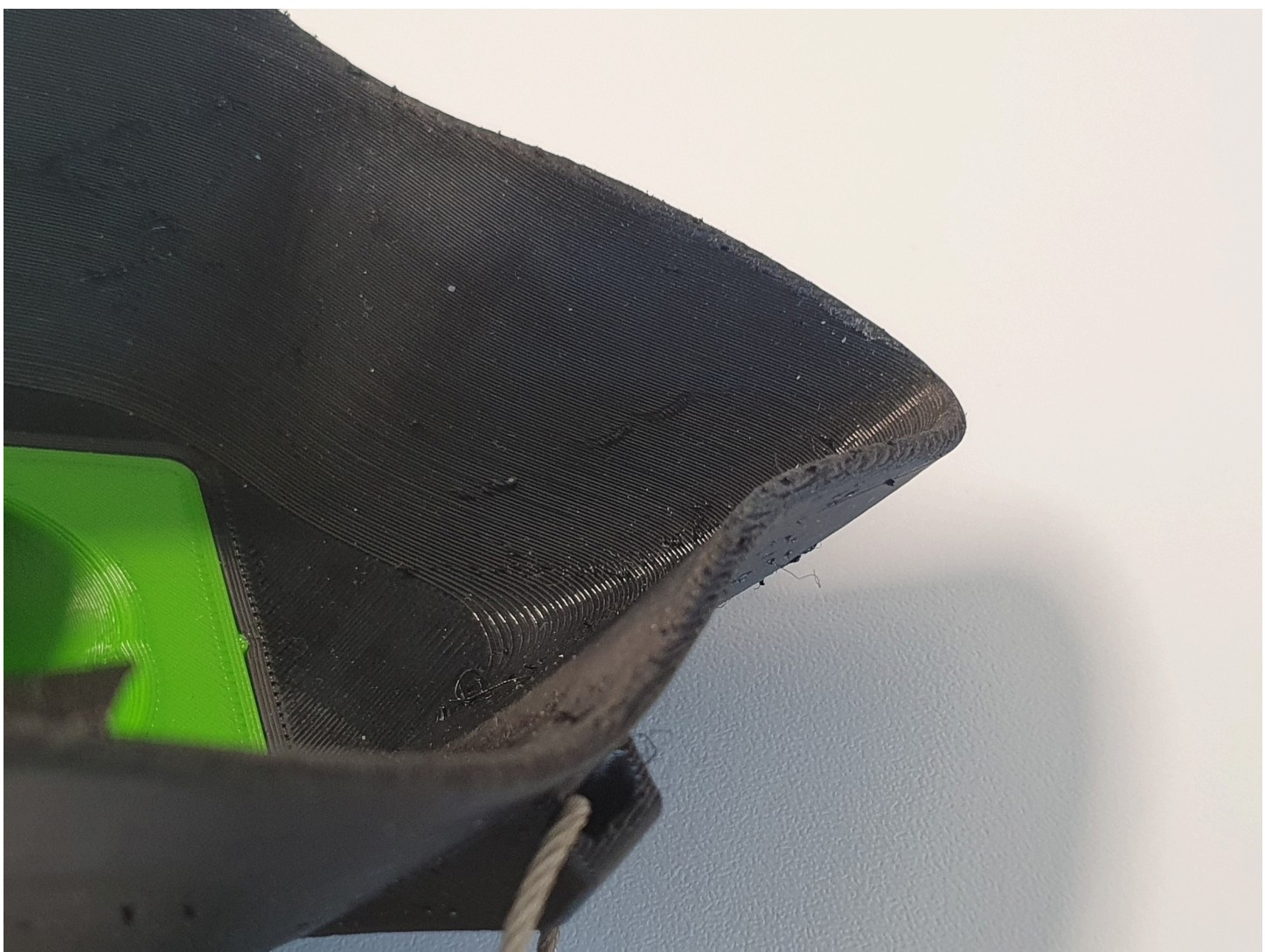

**Fig 6. Edge of the printed mask.** The edge of the printed mask is made of the same material as the mask itself and therefore relatively thin.

filament (Filamentworld, Neu-Ulm, Germany, 1.75mm diameter) and with 205˚C nozzle temperature. It was attached to two HME-filters (Iso-Gard #19212, Teleflex Medical 116 GmbH, Fellbach, Germany) and tightly strapped to the head (Fig 7).

All masks were measured 3 times in a row, the 3D-printed mask 4 times.

## Statistics

Results from the gamma camera were analyzed with the SPSS software package version 26 (IBM, Armonk, NY, USA). We used ANOVA for multiple comparisons. Post hoc analysis was done by means of the LSD test. A $p < 0.05$ was considered to be significant.

## Results

The home-sewn mask from a vacuum cleaner-bag had a filter efficacy of 69.76 ± 1.63%. It took 20min to sew it with a standard sewing-machine and moderate skills, the expiration valve was

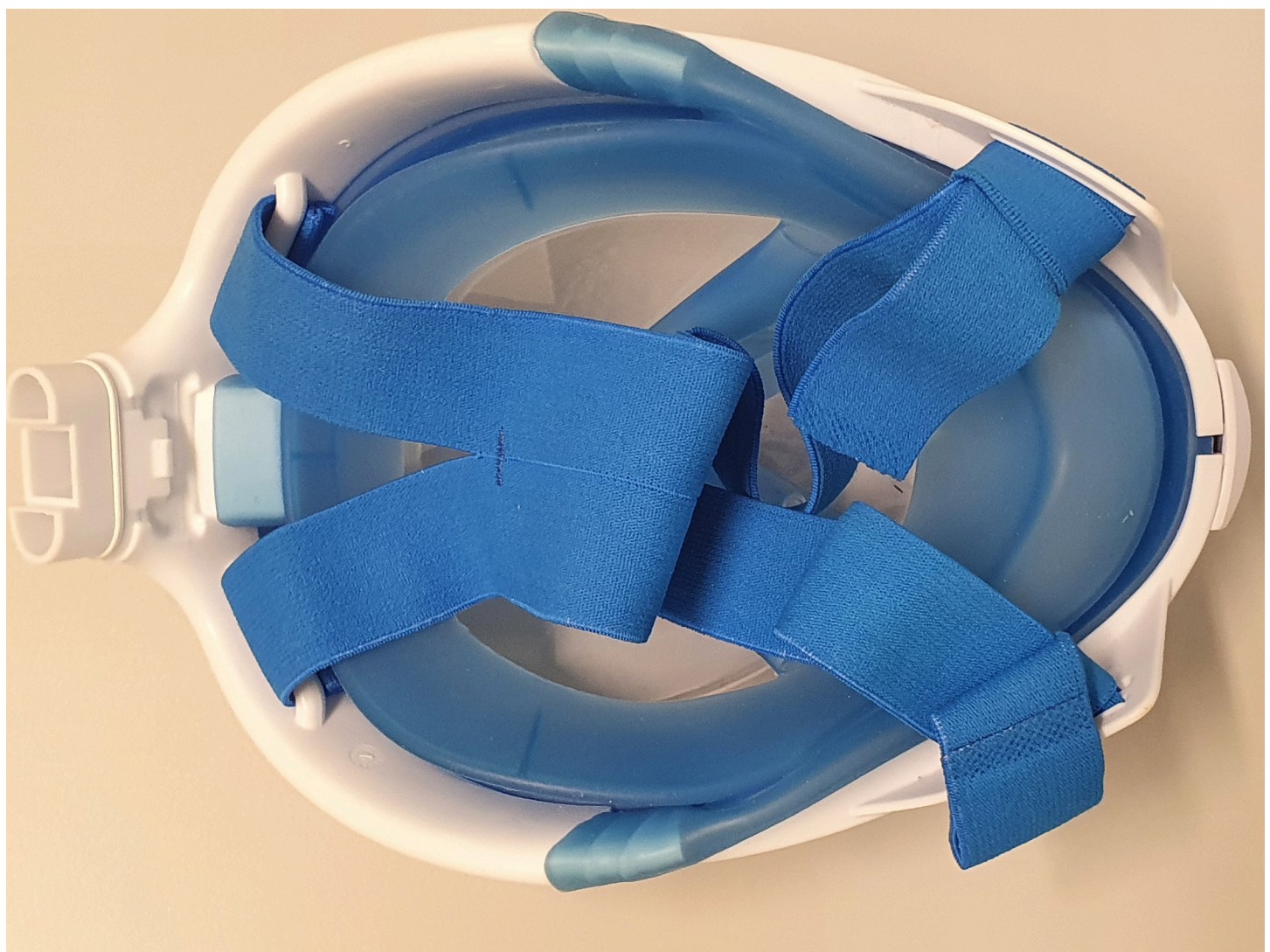

**Fig 7. Face-side of commercial mask.** The edge (blue rubber) of the commercial mask Easybreath⃝R is thick and made of a different material than the mask. It therefore creates a tight sealing.

printed in 20min. The 3D-printed solid face mask took 291min (mask) + 66min (adapter) to be produced on a 3D-printer, its efficacy was at 39.27 ± 2.08%. The Easybreath ⃝R diving mask was connected to two filters with an adapter which was printed within 160min. The mask had an efficacy of 85.07 ± 4.18% (Table 1). The filtration efficacy of all masks was significantly different (p<0.0001, Fig 8).

## Discussion

Respiratory protective devices in the European Union (EU) are certified using the EN149:2001 standard [9]. Medical masks instead are certified using the EN14683:2005 standard, of which the IR and IIR classes are splash-resistant [10]. However, the difference of these standards is that EN149:2001 demands filtration of air-born water- or oil-soluble particles and viruses, while EN14683:2005 requires resistance against direct splashes. Certification with EN149:2001 has to be done by notified bodies, while EN14683:2005 can be classified by the manufacturer.

**Table 1. Results of the measurements.** This table shows the results of the measurements of the three home-made and modified masks.

| Setting | Test-Run | Activity Background | Activity Filter Product | Activity Filter Reference | Passthrough | Efficacy (run) | Efficacy (mean) |
|---|---|---|---|---|---|---|---|
| Mask 1 | 1 | 187 | 3567 | 5846.36 | 0.61 | 0.39 | 39.27 |
| | 2 | 190 | 4498 | 7090.00 | 0.63 | 0.37 | |
| | 3 | 222 | 4019 | 6702.73 | 0.60 | 0.40 | |
| | 4 | 286 | 3692 | 6310.91 | 0.59 | 0.41 | |
| Easybreath ® | 1 | 453 | 376 | 3464.55 | 0.11 | 0.89 | 85.07 |
| | 2 | 456 | 395 | 2680.91 | 0.15 | 0.85 | |
| | 3 | 408 | 920 | 4790.91 | 0.19 | 0.81 | |
| Vacuumcleanerbag | 1 | 536 | 2458 | 7671.82 | 0.32 | 0.68 | 69.76 |
| | 2 | 530 | 1824 | 6322.73 | 0.29 | 0.71 | |
| | 3 | 470 | 2463 | 8252.73 | 0.30 | 0.70 | |

As air-borne aerosols are diffuse, EN149:2001-masks have fit tightly to the face, while EN14683:2005 protect against direct splashes and need no tight sealing at the sides. The two type of masks recommended as PPE in the corona-pandemic are FFP2 and FFP3. FFP2-masks may have a mean-leakage of maximum 8% and a protective of at least 95% against a standard formula. FFP3-device have a maximum allowed mean-leakage of 2% and a protection of at least 99% against standard formula. These two European standards are comparable with the US-standard NIOSH-42CFR84 allowing 95% efficiency for N95 and 99% efficiency for N99 masks [11] and the Chinese standard GB2626-2006 [12], which was also confirmed by a technical bulletin of the company 3M® [13]. In general, devices like respiratory masks have to be certified with the CE-sign before they can be used within the European Union. In the US, the Food and Drug Administration (FDA) is the respective authority. Facing the shortage of certified PPE during the Corona pandemic, the recommendation EU2020/403 of the European Commission has allowed to make non-CE marked respiratory masks available to medical professionals after an accelerated and temporary certification process [14]. Additionally, the usually fee-based norms of each participating country of the EU are now available for free [15] in order to foster the development of high-quality PPE.

Fused Deposition Modeling (FDM) is a three-dimensional additive printing technique which allows the rapid production of mostly plastic-based objects. It was invented and patented in the late 80ies by the printer-company Stratasys®. After the expiration of their key-patents in 2009 [16], 3D printers have become affordable and widely available to non-professional users. The RepRap-Movement was founded by Adrian Bowyer, fostering the development of self-replicating 3D-printers [17]. The increasing quality of 3D scanners and printers gives rise to the question whether reproduction of parts of commercial items infringe the respective copyright [18], thus leading to the discussion of a "right to repair" [19] in the European Union. During the Corona pandemic, such legal questions became very evident considering the acute lack of ventilators, masks, shields and even simple components like adapters and breathing tubes and its life-threatening consequences [20,21]. Additionally, home-made masks, shields and other equipment are neither medically nor in any other way certified devices. Although the MakerMask project just received an approval by NIH for "(. . .)use outside of the direct healthcare setting and benefits critical front-line essential service providers including: Police/Law Enforcement, Fire and Rescue and other Emergency Response service providers." [1], it is not a certified medical device of the FDA. The same holds true for face shields, e.g. from the Czech printer company Prusa Research®, which were verified by the Czech Ministry of Health and are currently used in some US-states, but are still not FDA/CE-marked medical devices [2]. Certification and ensuring a certain standard of quality achieved

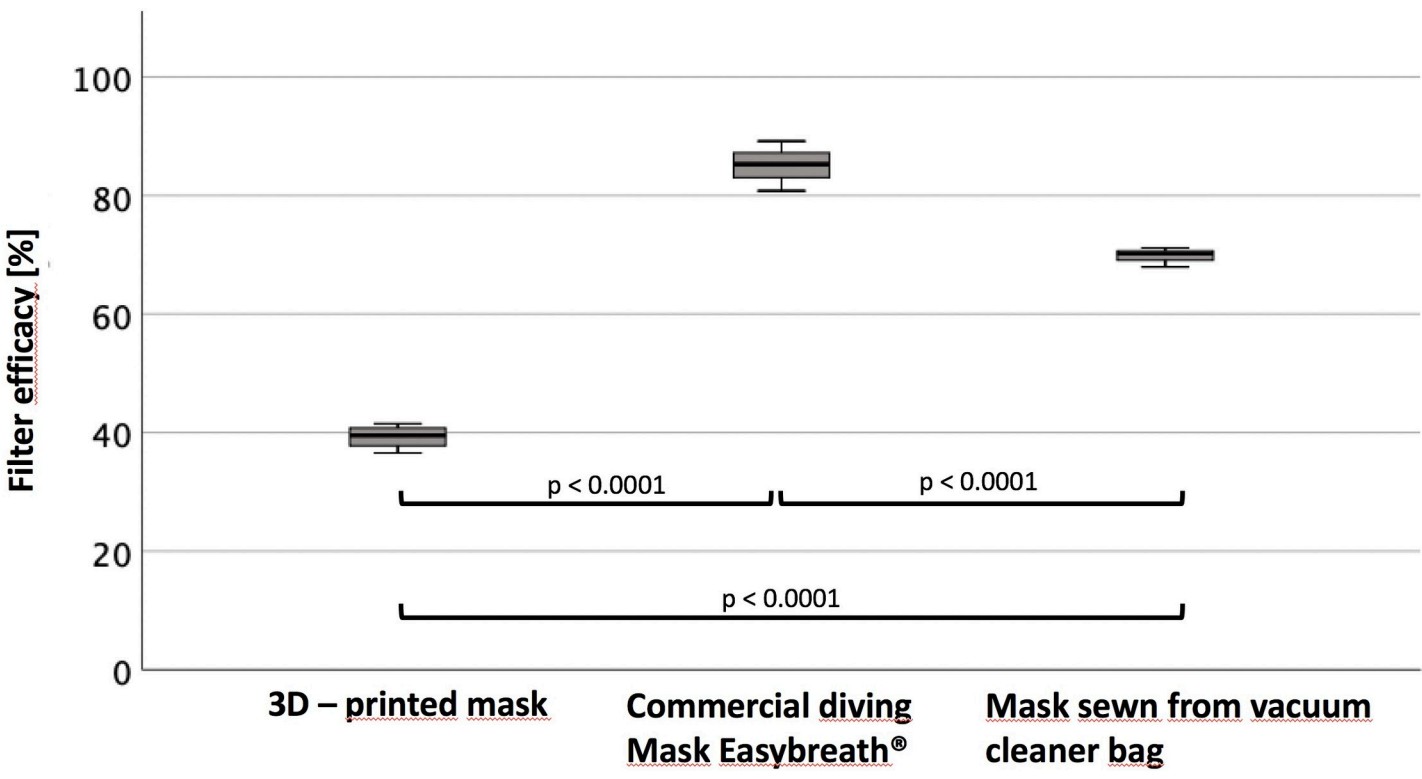

**Fig 8. Efficacy.** Filter efficacy of the respective filters and their statistically significant difference.

over decades of development and political process is still crucial for medical devices, even in difficult situations. The FDA allowed a 3D-printed adapter designed by the company Formlabs® to be certified in the fast-track certification process [22], and certification still remains an important step even in cases manufacturers are forced to produce different products than usual, e.g. Tesla ® being asked to produce ventilators instead of cars. Additionally, it has to be regulated how long non-certified "emergency products" shall be used, i.e. when the pandemic situation is officially over and from which point on only certified devices or PPE are allowed. In catastrophic situations, such legal aspects might be temporarily irrelevant, but they always harbor the risks of compensation claims. These claims might not be limited to actual infections of insufficient PPE, but could also extend to injuries which might have been caused by damaged material. Ultimately, it will be a legal task for courts to ascertain whether the use of non-certified products justifies compensation claims. In our personal experience (MG) the question arose who would be liable for injuries of broken plastic face shields.

The extent to which legal risks exist, depends on the legal system of the production site and the application site. A complete analysis of legal risks and all the more so of the applicable regulations would go far beyond the scope of this article.

In order to minimize legal risks, however, transparent product information is likely to be of the utmost importance, in addition to adequate quality assurance during production. If it is presented transparently to the user what the product's claim is, under which (regulatory) conditions it is manufactured and which requirements it may also not fulfil, it is up to the user to assess the risk of whether the product in question can and should be used.

According to European law, the intended purpose is of particular importance, i.e. for what purpose the manufacturer offers a product. If the product is manufactured for the purpose "to

be worn or held by a person as protection against one or more risks to his health or safety" (Art. 3, Reg. (EU) 2016/425), it is personal protective equipment in the sense of the Regulation (EU). If, on the other hand, the product is manufactured and offered for the purpose of disease prevention, the product falls under the Medical Device Directive (Reg. (EU) 2017/745) and is subject to the requirements set out therein. Although in the course of the corona pandemic—out of necessity—the requirements for medical devices and PPE were reduced, sometimes severely, (examples include the "automatic" recognition of PPE produced according to Chinese standards or the highly controversial reprocessing instructions of German ministries on surgical face masks and FFP), high standards still apply in this area, which must also be complied with unless official or legislative simplifications are expressly provided. For example, conformity assessment procedures must be carried out for both medical devices and PPE and the "essential requirements" required by the respective regulations must be observed and documented.

Any type of homemade protective equipment which does not meet the above-mentioned requirements can be used for self-protection, but placing it on the market, i.e. handing it over to third parties, appears to be highly problematic outside of emergency situations such as the corona crisis. But, even during the crisis, especially in particularly sensitive medical areas material has to be preferred that has undergone the intended testing and evaluation procedures, if that is possible.

Even in an emergency situation, however, the person providing assistance—in this case by providing home-made PPE–has to exercise due care. If the products are handed over free of charge, i.e. if a gift is involved, liability may be reduced in the form of a limitation of liability to intent and gross negligence (e.g. § 521 BGB (Germany)). Even the unselfish actor who gives away the self-made products is therefore liable for damages resulting from errors which the (here untechnical) manufacturer should have recognized. The requirements increase depending on the abilities of the manufacturer. The degree of negligence involved in making a community mask on a home sewing machine will therefore be different from that involved in making face-shields by a professional 3D printing company staffed with engineers, as the ability to detect unacceptable design flaws is simply more pronounced.

Despite these legal aspects, it was the aim of this small study to investigate the principle usefulness of selected home-made PPE.

Analog to the good efficiency of commercial continuous-positive-airway-pressure (CPAP)-masks attached to antiviral filters [6], the full-face diving mask Easybreath ® shows good filtration result. This mask has a separate inspiration and expiration pipe in order to reduce dead space. However, increases in CO2 levels cannot be excluded, especially if the mask is worn for a longer period and the user has a more rapid shallow breathing during physical activity or mental stress. Additionally, proper disinfection might be an issue if the masks are used in highly contaminated environments. This type of mask has been used in Italian hospitals for emergency CPAP-ventilation with the same adapter we used in our study, here attached to two isoguard-filters [23]. The whole process to convert the diving mask into a PPE was mostly attributed to the printing of the adapter to accommodate those filters, which was 160 minutes.

The printed face mask in the standard-size showed only 39.2% filtering efficiency. As the mask was equipped with the same antiviral filters as the Easybreath® mask, this lack of efficiency was most likely caused by a suboptimal fitting on the dummy head. However, also a smaller and a larger mask (printed 5% smaller and 10% larger than the original one) did not fit on the face at all with obvious visible gaps, even though the mask was printed not from hard plastic, but the soft rubber-like material TPE. In reality, a smooth skin and cheek fat might reduce this respiration bypass. The bad fitting is mostly contributed to the thin printed edge and the printing-lines, which is less flexible than e.g. the rubber of commercial diving masks.

However, the problem of sub-optimally fitting masks has already been addressed in a technical note published by Swennen et al. who used a smartphone based face-scanning app in order to produce tight-fitting, customized face masks [24]. Additionally, printed masks have to be inspected before use as small gaps might occur during printing, resulting in unintentional leakage. The problem of disinfection has also to be addressed in contagious environments. The printing of one mask and the respective adapter took 291 + 66 minutes. Adapting the masks to a face scan would even take longer, and it has to be questioned whether this time is really well-invested considering the poor overall filter-efficacy.

Using home-made 3D Printed template, it took 20 minutes to sew the 2 Layer mask from microfleece material used in a vacuum cleaner bag and 20 minutes to print the expiration valve. After the military university Munich published a study showing that in general vacuum cleaner bags retain aerosols very efficiently [25], numerous manufacturers of cleaner bags discouraged the public to sew masks with their material as it is beyond its intended use. The user has to make sure that there are no harmful chemicals such as glass fibres within the material of the bag before it is used for constructing a mask. However, in our experiment and in line with the results published by the military university Munich, the masks made of a cleaner bag had a efficacy below, but near to a commercial N95 mask in our previous experiments [6]. The one-way expiration valve is not necessary for the proper function of the mask and is only applied to make the longer use of the mask more comfortable. The flexible properties of the fleece created a good sealing on the mannequin face and therefore the mask shows a good efficacy. Considering the short production time of the mask and this good efficacy, home-sewn masks of such materials might be recommendable if the user assures that no harmful chemistry is in the material of the cleaner-bag.

## Conclusion

In the Corona pandemic, the lack of PPE puts medical personnel at risk. Therefore, home-made solutions for face shields and masks as well as other PPE are created by the 3D "maker" community and distributed among medical professionals. Some types of masks and modifications of commercial diving equipment, which are tested in this study, show good filtration efficacy, the 3D-printed face masks instead only have limited filter efficacy. However, it must be emphasized that none of presented solutions have medical clearance or certifications. The most important factor seems to be a tight fitting on the face and a good sealing at the nose and cheeks. Without tight fitting, the best filter cannot perform optimally. Despite the fact that catastrophic situations demand extraordinary solutions and that some tested home-made equipment shows excellent results, its use is at one's personal risk and legal aspects–at least in the end of the pandemic–should not be forgotten.

## Author Contributions

**Conceptualization:** Mortimer Gierthmuehlen, Bernd Kuhlenkoetter, Yaroslav Parpaley.

**Data curation:** Dominic Dellweg.

**Investigation:** Mortimer Gierthmuehlen, Stephan Gierthmuehlen, Dieter Köhler, Dominic Dellweg.

**Methodology:** Mortimer Gierthmuehlen, Dieter Köhler, Dominic Dellweg.

**Writing – original draft:** Mortimer Gierthmuehlen, Bernd Kuhlenkoetter, Yaroslav Parpaley, Stephan Gierthmuehlen, Dieter Köhler, Dominic Dellweg.

**Writing – review & editing:** Mortimer Gierthmuehlen, Yaroslav Parpaley.

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
