## [Decision Letter · Decision Letter 0]

25 Jun 2020

PONE-D-20-13641

Evaluation and discussion of handmade face-masks and commercial diving-equipment as personal protection in pandemic scenarios

PLOS ONE

Dear Dr. Gierthmuehlen,

Thank you for submitting your manuscript to PLOS ONE. After careful consideration, we feel that it has merit but does not fully meet PLOS ONE’s publication criteria as it currently stands. Therefore, we invite you to submit a revised version of the manuscript that addresses the points raised during the review process.

We look forward to receiving your revised manuscript.

Kind regards,

Chee Kong Chui, PhD

Academic Editor

PLOS ONE

Journal Requirements:

2. Please include your tables as part of your main manuscript and remove the individual files. Please note that supplementary tables (should remain/ be uploaded) as separate "supporting information" files

3. We note that Figures 3 and 4 in your submission contain copyrighted names (Philips Respironics. All PLOS content is published under the Creative Commons Attribution License (CC BY 4.0), which means that the manuscript, images, and Supporting Information files will be freely available online, and any third party is permitted to access, download, copy, distribute, and use these materials in any way, even commercially, with proper attribution. For more information, see our copyright guidelines: http://journals.plos.org/plosone/s/licenses-and-copyright.

1.    You may seek permission from the original copyright holder of Figures 3 and 4 to publish the content specifically under the CC BY 4.0 license.

'MG is founder and consultant of Neuroloop GmbH, which has no link to this manuscript. The authors therefore have declared that no competing interests exist.'.

Additional Editor Comments (if provided):

As an engineering paper, this paper is not quite technical. The discussion on the non technical matters, legal implication etc is good. Nevertheless, this work is relevant to the covid-19 pandemic. Please amend your paper according to the reviewers' comments.

Reviewers' comments:

Reviewer's Responses to Questions

**Comments to the Author**

1. Is the manuscript technically sound, and do the data support the conclusions?

Reviewer #1: Partly

Reviewer #2: Partly

2. Has the statistical analysis been performed appropriately and rigorously? 

Reviewer #1: Yes

Reviewer #2: Yes

3. Have the authors made all data underlying the findings in their manuscript fully available?

Reviewer #1: Yes

Reviewer #2: No

4. Is the manuscript presented in an intelligible fashion and written in standard English?

Reviewer #1: Yes

Reviewer #2: Yes

5. Review Comments to the Author

Reviewer #1: The manuscript reports a study on the filtration efficiency of 3 home-made or modified masks for use in the current Covid-19 pandemic. A large part of the manuscript involves a discussion on the complexities of the legal aspects in the manufacture and use of such home-made masks. Overall the manuscript is well written. However, there are some concerns that should be addressed:

1. With the large range of proposed designs and materials, the selection of the masks to be evaluated seems quite arbitrary. Perhaps the motivation or justification behind the selection of these 3 mask designs/models should be included in the text.

2. With respect to the vacuum cleaner bag mask – a ‘home-made design-template’ is mentioned with no reference from where its design was based on or how it was designed. Perhaps providing more details about it could assist readers in better understanding the manuscripts results and conclusions.

3. Please include images of the easybreath diving mask and its fit on the mannequin head. Similarly, please provide an image illustrating the fit of the 3D printed mask. This would assist in explaining the differences in filtration efficiency across the masks.

4. In the discussion, the poor filtration efficiency of the 3D printed mask was suggested to be due to the poor fitting of the mask. Would this be then an unfair comparison with respect to the vacuum cleaner bag and the diving mask? If they were poorly fitted as well, wouldn’t they result in similarly poor efficiency?

5. What does ‘can be efficient’ (in the abstract) and could the authors elaborate on how they came to this conclusion? Given that the filtration efficiency standards for EN149:2001 is >95% for FFP2-masks, this conclusion seems relatively weak, as all the tested masks seem to be relatively far from this threshold.

6. In the context of Covid-19, where transmission routes potentially involve droplets transporting the virus during coughing or sneezing, perhaps a more relevant standard would be that for surgical masks (EN14683). This standard involves more than just evaluating particle filtration efficacy and requirements such as bacterial filtration efficiency, breathability, splash resistance, biocompatibility and microbial cleanliness need to be met. While it is understandable that such strict requirements may be difficult for homemade mask makers to achieve, a discussion of this should be included for completeness.

7. The column ‘Setting’ in Table 1 should be translated to English or relabelled to reduce confusion, eg. ‘Staubsauger’ could be relabelled to ‘Mask sewn from vacuum cleaner bag’ as in Figure 5.

In summary the scientific contribution of this manuscript is limited. The discussion of the legal aspects of home-made masks is interesting but is again limited to European laws and appears to be incomplete. Nevertheless, any research helps in this global fight against the Covid-19 pandemic and the manuscript may provide an experimental foundation from which further studies can be performed.

Reviewer #2: The authors intended to study three different types of home-made masks effectiveness in response to the shortage of the masks amid the covid19 pandemic. The experimental design is reasonable. The filtration effectiveness was measured using scintigraphic camera. The cost and effectiveness of three types of home-made masks were compared. Legal aspects of these masks were thoroughly discussed. Though the paper does not directly propose a working solution to the shortage of masks amid pandemic, it gives the readers an idea about the role of 3D printers and their effectiveness and cost in printing masks.

multiple numerical typos, should be dot, instead of comma, e.g. 0.63 not 0,63.

line 78: It is unclear what the sentence 'with drying, we calculated a diameter of 0.63micro-meter for the NaCl-aerosol' means. How does the 0.63 relate with the 2.4-3.3 diameter mentioned in previous sentence?

Line 98: 10 breaths with 1L were performed over 50s. It is good to mention that a healthy adult's respiratory frequency and lung capacity so that the readers can compare it with the experimental setup.

line 103: Does the ratio of measured counts for filters depend on radioactivity? The radioactivity was counted for one minute. Since 99m-Tc has a half life of 6 hours, which is not much longer than the experimental time assumed to be the magnitude of hours. It is good to discuss the timeline as of when the radioactivity was counted to account for the decay.

line 120: the link is not working, after clicking, i was redirected to homepage showing that i am not authorized to view the page.

line123: link not working either.

line 272: due to the lack of tight-fitting of the printed plastic masks, the experimental results about effecacy maybe not trustworthy. The fitness of the mask to face might be the most important factor that matters. It is very interesting to see whether the smartphone based face scanning can greatly improve the effectiveness given same experimental design.

After all the experiments, it is good to make a tentative conclusion and recommendations to the readers and policy makers as of what is the best practice given current situation, rather than just describe what you found in the experiments.

6. PLOS authors have the option to publish the peer review history of their article (what does this mean?). If published, this will include your full peer review and any attached files.

Reviewer #1: No

Reviewer #2: No

---

## [Author Response · Author response to Decision Letter 0]

18 Jul 2020

Reply to the Editor

We thank the Editor for the comments on the formal appearance of our manuscript. We corrected the indicated points accordingly. 

Reply to the Reviewers

We also thank the Reviewers for their time to read and comment on our paper, especially in these extraordinary times. In the following we would like to reply to their comments and hope to meet their expectations.

Reviewer #1

The manuscript reports a study on the filtration efficiency of 3 home-made or modified masks for use in the current Covid-19 pandemic. A large part of the manuscript involves a discussion on the complexities of the legal aspects in the manufacture and use of such home-made masks. Overall the manuscript is well written. However, there are some concerns that should be addressed:

1. With the large range of proposed designs and materials, the selection of the masks to be evaluated seems quite arbitrary. Perhaps the motivation or justification behind the selection of these 3 mask designs/models should be included in the text.

This is correct. We added the information why we chose these specific masks and designs in the Introduction section. 

2. With respect to the vacuum cleaner bag mask – a ‘home-made design-template’ is mentioned with no reference from where its design was based on or how it was designed. Perhaps providing more details about it could assist readers in better understanding the manuscripts results and conclusions.

The design was created by the author YP with respect to general commercial designs. The template will be added as Fig. 4b.

3. Please include images of the easybreath diving mask and its fit on the mannequin head. Similarly, please provide an image illustrating the fit of the 3D printed mask. This would assist in explaining the differences in filtration efficiency across the masks.

The mannequin we used for this study has already been modified for another project and does therefore not represent the original situation anymore. We added close-up photographs of the edges of both the printed and the commercial diving mask to demonstrate the differences between the sharp edge of the printed mask (made of the same material as the mask itself) and the smooth edge of the easybreath made of rubber. 

4. In the discussion, the poor filtration efficiency of the 3D printed mask was suggested to be due to the poor fitting of the mask. Would this be then an unfair comparison with respect to the vacuum cleaner bag and the diving mask? If they were poorly fitted as well, wouldn’t they result in similarly poor efficiency?

That comment is true. However, the commercial mask has a rubber-edge which tightly fits to the face, and there are only 3 sizes available. As the rubber is flexible and wide it seals very well. We had 3 sizes for the 3D-printed masks as well, but none of them fitted well. This is contributed to the different materials (rubber vs. flexible filament) and the thickness of the edges (wide vs. thin, smooth surface vs. printed surface with visible lines). We stated that, in reality with a real face, the subcutaneous fat of the cheeks might create a better sealing, but this is hard to investigate with a standard mannequin. 

The vacuum cleaner consists of t a very flexible woven material which fits tightly to the face. Even though YP and MG have different faces, the masks create a good sealing even when one size is used on different faces. It is therefore all about flexibility of the material. 

5. What does ‘can be efficient’ (in the abstract) and could the authors elaborate on how they came to this conclusion? Given that the filtration efficiency standards for EN149:2001 is >95% for FFP2-masks, this conclusion seems relatively weak, as all the tested masks seem to be relatively far from this threshold.

This is correct, we changed the respective sentence in the abstract. For us it seemed interesting to show the differences between several designs tested on a single mannequin head. 

6. In the context of Covid-19, where transmission routes potentially involve droplets transporting the virus during coughing or sneezing, perhaps a more relevant standard would be that for surgical masks (EN14683). This standard involves more than just evaluating particle filtration efficacy and requirements such as bacterial filtration efficiency, breathability, splash resistance, biocompatibility and microbial cleanliness need to be met. While it is understandable that such strict requirements may be difficult for homemade mask makers to achieve, a discussion of this should be included for completeness.

We included a paragraph discussing the mentioned standard. 

7. The column ‘Setting’ in Table 1 should be translated to English or relabeled to reduce confusion, eg. ‘Staubsauger’ could be relabeled to ‘Mask sewn from vacuum cleaner bag’ as in Figure 5.

We reformatted the whole table and inserted it into the manuscript. 

In summary the scientific contribution of this manuscript is limited. The discussion of the legal aspects of home-made masks is interesting but is again limited to European laws and appears to be incomplete. Nevertheless, any research helps in this global fight against the Covid-19 pandemic and the manuscript may provide an experimental foundation from which further studies can be performed.

Reviewer #2 

The authors intended to study three different types of home-made masks effectiveness in response to the shortage of the masks amid the covid19 pandemic. The experimental design is reasonable. The filtration effectiveness was measured using scintigraphic camera. The cost and effectiveness of three types of home-made masks were compared. Legal aspects of these masks were thoroughly discussed. Though the paper does not directly propose a working solution to the shortage of masks amid pandemic, it gives the readers an idea about the role of 3D printers and their effectiveness and cost in printing masks.

Multiple numerical typos, should be dot, instead of comma, e.g. 0.63 not 0,63.

We corrected that accordingly.

line 78: It is unclear what the sentence 'with drying, we calculated a diameter of 0.63micro-meter for the NaCl-aerosol' means. How does the 0.63 relate with the 2.4-3.3 diameter mentioned in previous sentence?

The diameter of 2.4-3.3µm relates to the diameter in “wet” conditions and a concentration in 0.9% NaCl. With drying and calculated to a concentration of 1 the diameter of the particles is 0.58 - 0.63µm. We corrected this value accordingly. This is a standard-procedure in aerosole-physics, the respective formula is:

mass= concentration x volume

concentration1=0.9% NaCl=0.009

MMAD=2.8µm=d1

r1=1.4µm

volume1=4/3 x � x 〖1.4µm〗^3=11,49µm^3

concentration2 (dryed particles)=1

concentration1 x volume1=concentration2 x volume2

concentration1 x volume1=1x volume2

0,009 x 11,49µm^3=volume2=0,10µm^3

r2=0,29µm

d2=0,58µm

Line 98: 10 breaths with 1L were performed over 50s. It is good to mention that a healthy adult's respiratory frequency and lung capacity so that the readers can compare it with the experimental setup.

We inserted the respective values as suggested. 

line 103: Does the ratio of measured counts for filters depend on radioactivity? The radioactivity was counted for one minute. Since 99m-Tc has a half-life of 6 hours, which is not much longer than the experimental time assumed to be the magnitude of hours. It is good to discuss the timeline as of when the radioactivity was counted to account for the decay.

Both test-filter and the reference were exposed to Tc99m at the very same time and the entire measurements of all masks took appr. 1h. We used a new reference filter every time. That is why the natural decay of Tc99m seemed irrelevant. The measurement under the gamma-camera was done right after the ventilation with the artificial lung, the gamma-camera was in the same room as the test-setup.

line 120: the link is not working, after clicking, i was redirected to homepage showing that i am not authorized to view the page.

This is astonishing, the service Cults3D seems to be entirely down. The same template is accessible on Thingiverse, another public repository. We inserted the respective link and apologize for the inconvenience. 

line123: link not working either.

Same here, the template is available on Thingiverse.

line 272: due to the lack of tight-fitting of the printed plastic masks, the experimental results about effecacy maybe not trustworthy. The fitness of the mask to face might be the most important factor that matters. It is very interesting to see whether the smartphone based face scanning can greatly improve the effectiveness given same experimental design.

Yes, that will definitely be an interesting future study we might plan in the near future. 

After all the experiments, it is good to make a tentative conclusion and recommendations to the readers and policy makers as of what is the best practice given current situation, rather than just describe what you found in the experiments.

We added a respective sentence, though we cannot officially recommend any solution primarily to legal aspects.

---

## [Editor Report · Decision Letter 1]

6 Aug 2020

Evaluation and discussion of handmade face-masks and commercial diving-equipment as personal protection in pandemic scenarios

PONE-D-20-13641R1

Dear Dr. Gierthmuehlen,

We’re pleased to inform you that your manuscript has been judged scientifically suitable for publication and will be formally accepted for publication once it meets all outstanding technical requirements.

Kind regards,

Chee Kong Chui, PhD

Academic Editor

PLOS ONE

Additional Editor Comments (optional):

The authors have adequately addressed the comments and issues raised by the reviewers. I recommend that this paper be accepted for publication. It is an interesting paper amidst the coronavirus outbreak.
---

## [Editor Report · Acceptance letter]

10 Aug 2020

PONE-D-20-13641R1 

Evaluation and discussion of handmade face-masks and commercial diving-equipment as personal protection in pandemic scenarios 

Dear Dr. Gierthmuehlen:

I'm pleased to inform you that your manuscript has been deemed suitable for publication in PLOS ONE. Congratulations! Your manuscript is now with our production department. 

Kind regards, 

on behalf of

Dr. Chee Kong Chui 

Academic Editor

PLOS ONE